# Idiopathic Multicentric Castleman Disease with Strikingly Elevated IgG4 Concentration in the Serum and Abundant IgG4-Positive Cells in the Tissue: A Case Report

**DOI:** 10.3390/diagnostics12092261

**Published:** 2022-09-19

**Authors:** Chia-Chun Cheng, Ying-Chu Chen, Yung-Hsiang Hsu, Kuei-Ying Su

**Affiliations:** 1Department of Pathology, Hualien Tzu Chi Hospital, Buddhist Tzu Chi Medical Foundation, Hualien 970, Taiwan; 2Department of Physical Medicine and Rehabilitation, Hualien Tzu Chi Hospital, Buddhist Tzu Chi Medical Foundation, Hualien 970, Taiwan; 3Division of Allergy, Immunology and Rheumatology, Hualien Tzu Chi Hospital, Buddhist Tzu Chi Medical Foundation, Hualien 970, Taiwan; 4School of Medicine, Tzu Chi University, Hualien 970, Taiwan

**Keywords:** idiopathic multicentric Castleman disease (iMCD), IgG4-related disease (IgG4RD), multiple lymphadenopathies

## Abstract

Idiopathic multicentric Castleman disease (iMCD) can be challenging to distinguish clinically and histopathologically from Immunoglobulin G4-related disease (IgG4RD). A 73-year-old man was referred to a rheumatologist for suspected autoimmune-related polyclonal hypergammaglobulinemia. The patient had a history of multiple lymphadenopathies in the neck for over 20 years. Laboratory data showed elevated serum immunoglobulin G4 (IgG4) levels, hypergammaglobulinemia, high C-reactive protein (CRP) levels, marked anemia, and positivity for several autoantibodies. Additionally, imaging studies revealed multiple enlarged lymph nodes and multifocal, ill-defined, small patchy opacities over the lung. Biopsies of the neck lymph node and right lung revealed typical features of multicentric Castleman disease (MCD). Immunohistochemical staining was negative for human herpesvirus-8 (HHV-8) in both lymph nodes and the right lung, sub-classified as iMCD, whereas the IgG4/IgG ratio was >40%, which raised the suspicion of IgG4RD. However, serological cytokine analysis demonstrated an increased interleukin-6 (IL-6) level, alongside systemic inflammatory and histopathological features, distinguishing MCD from IgG4RD in this patient. The patient was treated with short-term glucocorticoids and regular infusion of an anti-IL-6 receptor monoclonal antibody (tocilizumab), with satisfactory clinical and radiographic responses. Notably, differentiating MCD from IgG4RD is crucial for optimal treatment. Clinical and pathological features may assist in distinguishing between these two diseases.

## 1. Introduction

Multicentric Castleman disease (MCD) is a rare lymphoproliferative disorder characterized by multiple lymphadenopathies [1], systemic inflammation, polyclonal lymphoproliferation, cytopenia, and multi-organ dysfunction. MCD is potentially life-threatening due to cytokine storm [2]. MCD can be further classified as classical MCD or idiopathic MCD (iMCD) based on the presence of human herpesvirus-8 (HHV-8). A subset of patients who meet the criteria for polyneuropathy, organomegaly, endocrinopathy, monoclonal gammopathy, and skin changes (POEMS) syndrome should also be identified before the diagnosis of iMCD is made for subsequent therapeutic strategies.

Other lymphoproliferative disorders and diseases may show clinically and histopathologically similar signs as MCD. This includes Immunoglobulin G4-related disease (IgG4RD). IgG4RD is a fibroinflammatory condition that affects almost every organ in the body, resulting in organ enlargement. Both diseases share laboratory findings, such as hyperglobulinemia and multi-organ dysfunction. Histopathologically, lymphoplasmacytic cells may be arranged in layers in thickened mantle zones, creating onion skin appearances in MCD, and mimicking a fibrotic reaction in IgG4RD. Furthermore, sheets of plasma cells in the interfollicular zone of the lymph node are a characteristic finding in plasma cells and mixed-type iMCD, which may also be easily confused with IgG4RD. However, several clinical and pathological features may help physicians and pathologists distinguish between iMCD and IgG4RD. This report shows a unique case of iMCD that was difficult to distinguish from IgG4RD because of strikingly elevated IgG4 serum concentration and abundant IgG4-positive cells in the tissue.

## 2. Case Presentation

A 73-year-old man had a history of neck lymphadenopathy for over 20 years and developed anemia and renal insufficiency over the past 6 years. He performed activities of daily living without fever, weight loss, edema, or arthralgia, except for severe soreness of the legs over the past 2 months. The patient was referred to a hematologist for abruptly worsening anemia (hemoglobin, 4.7 g/dL; reference range, 13.0–18.0 g/dL) and renal function (serum creatinine, 1.6 mg/dL; reference range, 0.7–1.2 mg/dL). Abnormal laboratory results included anemia (hemoglobin, 6.2 g/dL), hypoalbuminemia with albumin/globulin (A/G) reverse (albumin, 2.8 g/dL; reference range, 3.4–5.3 g/dL; globulin, 7.9 g/dL; reference range, 1.8–3.2 g/dL), abnormal serum levels of immunoglobulins (IgG, 5760 mg/dL; reference range, 700–1600 mg/dL; IgA, 746 mg/dL; reference range, 70–400 mg/dL; IgM, 586 mg/dL; reference range, 40–230 mg/dL; β2-microglobulin, 5998 ng/mL; reference range, 800–2340 ng/mL; free kappa (κ) light chain, 274 mg/dL; reference range, 3.3–19.4 mg/dL; free lambda (λ) light chain, 157 mg/dL; reference range, 5.7–26.3 mg/dL; and κ/λ, 274/157 = 1.75; reference range, 0.26–1.65), and increased inflammatory markers (C-reactive protein (CRP), 4.49 mg/dL; reference range, <0.5 mg/dL; erythrocyte sedimentation rate, >150 mm/hour; reference range, 0–20 mm/hour); pertinent negative results were noted in white cell count (6390/µL; reference range, 3900–10,600/µL), platelet count (397,000/µL; reference range, 150,000–400,000/µL), hepatic profiles (aspartate aminotransferase, 12 IU/L; reference range, <34 IU/L; alanine aminotransferase, 7 IU/L; reference range, <36 IU/L; total bilirubin, 0.5 mg/dL; reference range, 0.2–1.4 mg/dL; and direct bilirubin, 0.2 mg/dL; reference range, <0.4 mg/dL), and electrolytes (sodium, 136 mmol/L; reference range, 134–148 mmol/L; potassium, 4.1 mmol/L; reference range, 3.6–5.0 mmol/L; and calcium, 2.02 mmol/L; reference range, 2.0–2.5 mmol/L). Multiple myeloma was initially suspected; however, a subsequent bone marrow biopsy showed reactive plasma cell proliferation. Immunofixation electrophoresis of the bone marrow showed polyclonal gammopathy. Thus, multiple myeloma was unlikely. Surveys for chronic inflammatory disorders, autoimmune diseases, and infections were subsequently performed.

The workups for infectious diseases showed negative results for syphilis (Rapid Plasma Reagin non-reactive), human immunodeficiency virus (HIV) (Antigen/Antibody negative), active cytomegalovirus infection (CMV) (immunoglobulin M negative), tuberculosis (TB) (antigen, 0.18 IU/mL; reference range, 0–10 IU/mL), and Epstein-Barr virus infection (EB) (viral capsid antigen immunoglobulin M negative). Further workups for autoimmune diseases were positive for rheumatoid factor (RF) (30.3 IU/mL; reference range, 0–20 IU/mL), anti-cyclic citrullinated peptide antibody (ACPA) (43.7 EU/mL; reference range, 0–20 IU/mL), low positive of antinuclear antibodies (nucleolar, 1:40; reference range, <1:40), positive Coombs’ test, and negative for anti-neutrophil cytoplasmic antibodies and anti-double-stranded DNA antibodies. Notably, his serum IgG4 level was markedly elevated (1460 mg/dL; reference range, 10–140 mg/dL). The patient was referred to our rheumatology clinic for polyclonal gammopathy and positive RF and ACPA.

Physical examination showed a pale appearance and multiple lymphadenopathies at the neck and bilateral axilla, without active synovitis or skin lesions. Computed tomography of the head, neck, and chest showed multiple lymphadenopathies over the bilateral neck (Figure 1a) and mediastinum and multifocal ill-defined small patchy opacities over perihilar regions and the right middle lobe of the lung (Figure 1b).

Biopsies of the neck lymph node and lung nodule were performed. The neck lymph node biopsy showed sheets of plasma cells in interfollicular zones, atrophic follicles with a penetrating vessel, and a surrounding onion-skin-like mantle zone, creating a “lollipop” appearance. This picture is exclusive to Castleman disease (Figure 2a,b). A lung biopsy revealed massive plasma cells infiltrating the parenchyma (Figure 3a,b). No intranuclear immunoglobulin (Dutcher bodies) was found in the plasma cells; therefore, plasmacytoma was unlikely to be present. Furthermore, the absence of fibroblasts, myofibroblastic cells, and a monotonous population of abnormal lymphocytes excluded solitary fibrous tumors, inflammatory pseudotumors, and other neoplastic lymphoproliferative disorders such as follicular lymphoma. Subsequent laboratory data revealed elevated interleukin-6 levels (IL-6) (18.39 pg/mL; reference range, <7 pg/mL), indicating a systemic inflammatory status. Based on clinical, serological, and pathological findings, the patient was diagnosed with MCD. Although the bone marrow biopsy was non-diagnostic, the picture was compatible with MCD with bone marrow involvement [3].

Interestingly, the immunochemical analysis showed plasmacytic infiltration and IgG4/IgG ratio > 40% in the lymph nodes (Figure 4a,b) and lung tissue. The markedly elevated serum IgG4 level raised suspicion of IgG4RD. Nevertheless, typical histopathological characteristics of IgG4RD, such as storiform fibrosis and obliterative phlebitis, were not observed.

A further survey of MCD classification was performed. There was no polyneuropathy, organomegaly, endocrinopathy, monoclonal protein, or skin changes, excluding POEMS-associated MCD. Negative immunohistological staining for human herpesvirus-8 (HHV-8) excluded HHV-8-associated MCD. iMCD was diagnosed. Furthermore, TAFRO syndrome, a distinct systemic inflammatory disorder resembling several pathological features of iMCD, was excluded based on the 2019 updated diagnostic criteria and disease severity classification for TAFRO syndrome [4].

With anemia and pulmonary involvement, our patient fulfilled the criteria for severe iMCD [5]. He was treated fortnightly with tocilizumab, a humanized anti-IL-6 receptor monoclonal antibody, at 8 mg/kg intravenously. An initial high dose of glucocorticoids was prescribed and gradually tapered off in the subsequent 5 months. The patient has been regularly followed-up at our rheumatology clinic for 4 years with clinical improvements (hemoglobin, 12.7 g/dL; albumin, 4.1 g/dL; CRP, <0.1 mg/dL; and IgG4, 445 mg/dL) and shrinkage of lymph nodes (Figure 5a), where the size of lung mass modestly reduced from 3.11 cm to 1.69 cm (Figure 5b).

## 3. Discussion

Despite the presence of autoantibodies and markedly increased IgG4 levels in both serum and tissue, the characteristic manifestations of iMCD were evident in our patient, including multiple enlarged lymph nodes, a pulmonary lymphoplasmacytic mass, polyclonal hypergammaglobulinemia, anemia, renal dysfunction, and increased CRP and serum IL-6 levels. iMCD is a systemic inflammatory disorder driven by IL-6, which is responsible for systemic inflammation and B-cell activation [2,6]. The causative agent for IL-6 overproduction in classical MCD is HHV-8 infected plasmablasts/plasmacytes, but it remains unknown in iMCD. Common symptoms of iMCD include lymphadenopathy, fever, weight loss, anemia, renal insufficiency, and cytokine storms. However, the clinical course varies between patients. This patient experienced chronic lymphadenopathy for 20 years and chronic anemia for 5 years before the abrupt worsening of his clinical condition.

Otani et al. reviewed 22 cases of patients with iMCD and reported that the lungs were one of the affected organs in 15 cases (68%) [7]. The typical pulmonary manifestations of iMCD include pleural effusion, pulmonary edema, and lymphocytic interstitial pneumonitis [2]. Liu et al. reviewed 84 articles involving 128 patients and reported a significant proportion of pleural effusions (23%) and edema, ascites, or anasarca (23%) [8]. Pulmonary involvement is often observed in patients with iMCD. However, in our case, a solitary lung mass was found as an initial presentation, which is an unusual pulmonary manifestation in iMCD.

Until recently, the diagnosis of iMCD has been challenging due to the lack of standard diagnostic criteria. Fajgenbaum et al. reviewed data from 244 cases and established consensus diagnostic criteria that required both major (characteristic lymph node histopathology and multicentric lymphadenopathy) and 2 of 11 minor criteria (with at least one laboratory abnormality) [2]. These consensus diagnostic criteria facilitate prompt diagnosis, appropriate treatment, and collaborative research. Characteristic histopathological features include a constellation of regressed or hyperplastic germinal centers, follicular dendritic cell prominence, hypervascularization, and polytypic plasmacytosis [2]. Conversely, laboratory and minor clinical criteria include elevated CRP or erythrocyte sedimentation rate, anemia, thrombocytopenia or thrombocytosis, hypoalbuminemia, renal dysfunction or proteinuria, polyclonal hypergammaglobulinemia, constitutional symptoms, hepatosplenomegaly, effusions or edema, eruptive cherry hemangiomatosis or violaceous papules, and lymphocytic interstitial pneumonitis [8]. Importantly, iMCD mimics, such as infectious, malignancies, and autoimmune and autoinflammatory diseases, should be excluded before diagnosing iMCD.

In contrast, IgG4RD is a fibrous-inflammatory process that often leads to clinical enlargement of affected organs [9]. The most commonly affected areas are the pancreas, bile duct, major salivary and lacrimal glands, retroperitoneum, and lymphatic ducts [9,10]. Serologically, patients with IgG4RD often present with elevated serum IgG4 and IgE levels, hypergammaglobulinemia, and eosinophilia [9]. Typical histopathological characteristics include dense lymphocytic infiltration, positive immunostaining of IgG4-positive cells, storiform fibrosis, and obliterative phlebitis [9]. However, storiform fibrosis and obliterative phlebitis are less frequently detected in some organs. The prognosis varies based on the organ involved and the development of fibrosis [9]. Because of the diverse presentations, clinical conditions should be excluded before diagnosing IgG4RD based on the 2019 diagnostic criteria [10].

iMCD and IgG4RD are virtually exclusive, but may be challenging to distinguish on some occasions [11]. Both diseases can affect multiple organs, including the lymph nodes, lungs, and kidneys, and show hyperglobulinemia and increased levels of serum inflammatory biomarkers. Nevertheless, several clinical manifestations distinguish them. First, the presence of hepatosplenomegaly is more suggestive of iMCD, whereas involvement of orbits, lacrimal glands, salivary glands, or pancreas are manifestations of IgG4RD [7,12]. Next, age also appears to be a parameter due to the distinctive disease course (i.e., IgG4RD is more indolent than iMCD). Otani et al. compared several clinical features between iMCD and IgG4RD and noticed that iMCD (mean age at diagnosis: 53 ± 13 years) affected younger patients than IgG4RD (mean age at diagnosis: 69 ± 10 years, *p* < 0.001) [7]. The pathogenesis of increased IgG4 level in iMCD remains unclear [13]. Although the elevation of serum IgG4 levels alone could not distinguish between the two diseases, the serum IgG4/IgG ratio is a more reliable discriminator, with a mean value of 8.6% (1.5–23.6%) and 24.1% (8.7–59.0%) in iMCD and IgG4RD, respectively [7]. Additionally, an atopic history, including allergic rhinitis, asthma, and atopic dermatitis, is characteristic for IgG4RD [12]. Finally, several studies showed that elevated levels of IL-6, IgG, IgA, and IgM, increased CRP, anemia, and hypoalbuminemia are distinctive for iMCD [12,14,15]. The persistently elevated levels of serum CRP (≧1.0 mg/dL), IgA, or IgM are proposed as exclusion criteria for IgG4RD by the Japanese Pathology Study Group of IgG4-related disease [16].

Histopathologically, plasma cells infiltrating in the lymph nodes and lungs are seen in both IgG4RD and iMCD. The difference is that plasma cells in IgG4RD are of variable maturational stage whereas sheets of plasma cells known as grade 3 plasmacytosis may be displayed iMCD [2,7]. The IgG4/IgG ratio in tissue is more relevant than absolute IgG4-positive cell counts in distinguishing iMCD from IgG4RD [7]. In addition, eosinophil infiltration, obliterative phlebitis, and storiform fibrosis are almost exclusive to IgG4RD [7]. Hyalinized fibrosis of the lung parenchyma constitutes 75% of iMCD cases with lung involvement and is not observed in IgG4RD [7].

Collectively, although our patient’s age (73 years) at diagnosis and the serum IgG4/IgG ratio (1460/5760 = 25.3%) suggested IgG4RD, the serum IL-6, IgA, CRP, hemoglobin, and albumin levels accompanied by detectable autoantibodies were strongly suggestive of iMCD. Pathological examinations revealed sheets of plasma cells in the right lung and neck lymph nodes, without eosinophil infiltration, obliterative phlebitis, or storiform fibrosis. Despite this, the diagnosis of iMCD with abundant IgG4-positive cells was made based both on histopathological findings, i.e., sheets of plasma cells in interfollicular zones, atrophic follicles with a penetrating vessel in each and surrounding onion-skin-like mantle zone, and serological findings, i.e., elevated IL-6, IgA, and CRP levels, anemia, and hypoalbuminemia.

IL-6 contributes to host defense against acute environmental stresses such as infectious diseases or traumas. Dysregulated continuous IL-6 production may result in various autoimmune and chronic inflammatory diseases, including iMCD [17]. IL-6 acts on hepatocytes to induce a broad spectrum of acute-phase proteins, such as CRP, serum amyloid A, and fibrinogen, thereby decreasing albumin and transferrin [17]. As a B-cell activator, IL-6 induces B-cell differentiation into immunoglobulin-producing cells, resulting in polyclonal gammopathies or autoantibodies [17]. Thus, various autoantibodies could be found in iMCD patients [18].

An IL-6 antagonist was approved for treating patients with iMCD [8,19,20]. In 2018, an international working group of experts reviewed 344 cases of patients with iMCD and reached an evidence-based consensus that the first step in determining treatment is assessing severity [20,21]. Severe cases, estimated to account for 10–20% of iMCD cases, should be promptly started on a high-dose glucocorticoid regimen with anti-IL-6 therapy [21]. In our patient, anemia (hemoglobin, 4.7 g/dL) and pulmonary involvement were compatible with severe iMCD. Tocilizumab at 8 mg/kg IV fortnight was promptly administered with a high dose of glucocorticoids, and the latter was gradually tapered off after 5 months. After 4 years of treatment, the clinical symptoms and the serum laboratory data improved significantly, including hemoglobin (12.7 g/dL), CRP (<0.1 mg/dL), and IgG (445 mg/dL) levels. The lung mass reduced from 3.11 cm to 1.69 cm.

## 4. Conclusions

In summary, we reported on an iMCD patient with unusually increased IgG4 levels in the serum and IgG4-positive cells in the tissue. Although the elevated concentration of serum IgG4, abundant IgG4-positive cell infiltration, and high ratio of IgG4/IgG helped diagnose IgG4RD, a more precise diagnosis was required. These findings may also present in iMCD patients. Therefore, clinicians and pathologists should work together toward obtaining a correct diagnosis since iMCD and IgG4 share clinical (lymphadenopathy and multi-organ involvement) and pathological (lymphoplasmacytic infiltration) features.

## Figures and Tables

**Figure 1 diagnostics-12-02261-f001:**
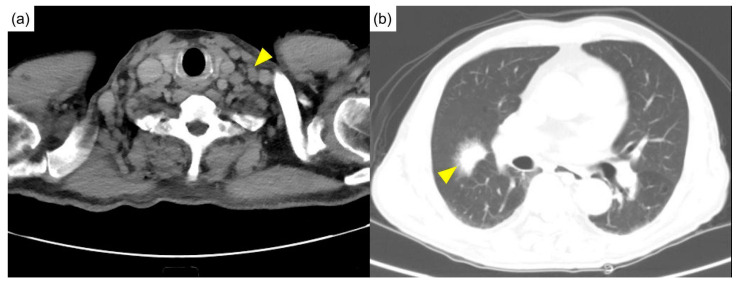
Initial computed tomography. (**a**) Diffuse lymphadenopathies (arrowhead) were noted in the neck computed tomography. (**b**) The chest computed tomography revealed a mass in the right middle lobe of the lung (arrowhead).

**Figure 2 diagnostics-12-02261-f002:**
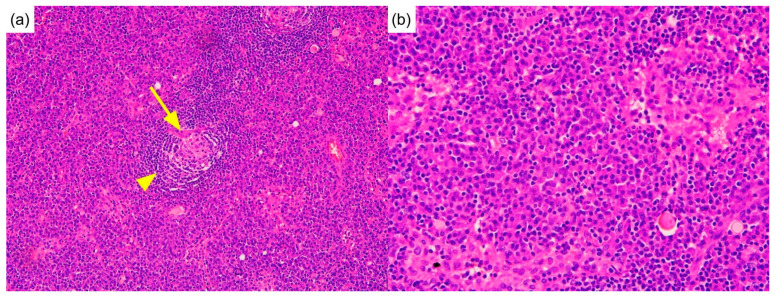
Lymph node biopsy. (**a**) Pathology of the lymph nodes shows an atrophic follicle with a penetrating vessel (arrow) and onion-skin-like mantle zone (arrowhead), creating a so-called “lollipop” appearance. (Hematoxylin and eosin, 200×) (**b**) Prominent plasma cells infiltrate the sinus. (Hematoxylin and eosin, 400×).

**Figure 3 diagnostics-12-02261-f003:**
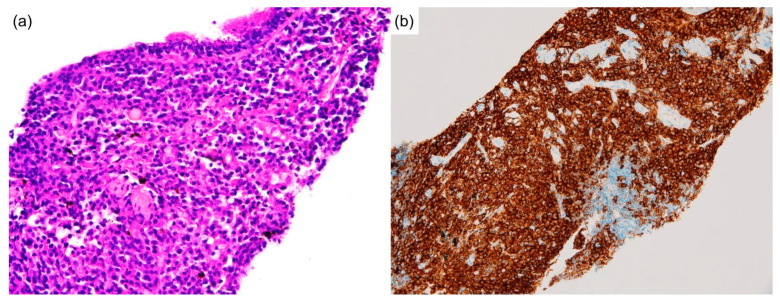
Lung biopsy, special staining. The lung biopsy shows (**a**) the parenchyma infiltrated by massive plasma cells, which could be stained by (**b**) CD138 immunohistologically. (Hematoxylin and eosin, and immunohistochemical stain, 100×).

**Figure 4 diagnostics-12-02261-f004:**
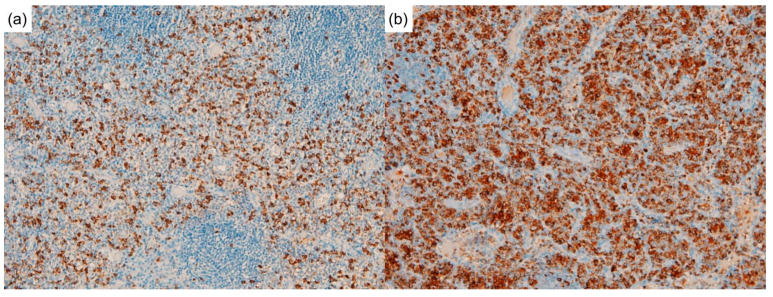
Lymph node biopsy, special staining. (**a**) immunoglobulin G4 (IgG4) stains (**b**) IgG stains. The IgG4 and IgG stains demonstrated an increased ratio of IgG4/IgG in the lymph node biopsy of more than 40% (100×).

**Figure 5 diagnostics-12-02261-f005:**
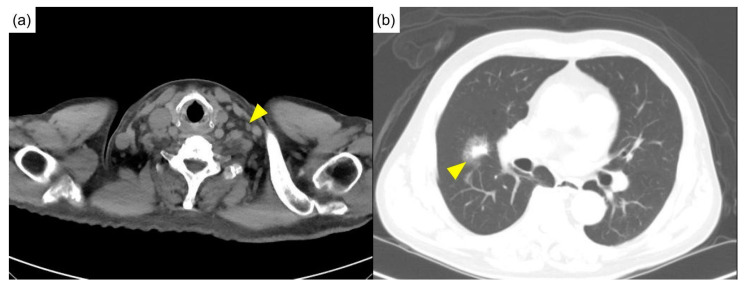
The latest computed tomography after 4 years of treatment. (**a**) Shrinkage of lymph nodes (arrowhead) was noted. (**b**) The size of the lung mass modestly reduced from 3.11 cm to 1.69 cm (arrowhead).

## Data Availability

The original contributions of this study are included in this article. Further inquiries can be directed to the corresponding authors.

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
