# Peer review of "Idiopathic Multicentric Castleman Disease with Strikingly Elevated IgG4 Concentration in the Serum and Abundant IgG4-Positive Cells in the Tissue: A Case Report"

_diagnostics, 2022, doi:10.3390/diagnostics12092261_

Round 1

Reviewer 1 Report

This is an interesting case report of a case of idiopathic MCD which compares the differences and similarities of the pathological, clinical, radiological, biochemical and serological features of iMCD with IgG4-RD.

The English needs to be edited by a native speaker as the expression is clunky at times (82- 83, 166-168, 201 eg). 

Normal range should be provided for all biochemistry and haematological parameters.

The discussion is somewhat repetitive and could be streamlined. The authors do not cite the large comparative study of Sasaki et al, 2017 on the same topic.  These authors note distinguishing features of organ involvement (orbits, lacrimal and salivary glands and pancreas in IgG4-RD), Atopic history in IgG4-RD, Increased levels of CRP and IgA in iMCD. 

Reviewer 2 Report

This reported case is unique and interesting, because multicentric Castleman’s disease (MCD) with IgG4-plasmacytosis. However, some revisions are necessary.

1. Author wrote “TAFRO syndrome , a subtype with the worst prognosis for iMCD” on page 4, however most researchers recently thought that TAFRO syndrome is different from typical iMCD (Am J Hematol 2019,94:975-983.). And some patients with TAFRO syndrome do not demonstrate lymphadenopathy. Therefore, iMCD and TAFRO syndrome may overlap in some parts of clinical and histological manifestation, but TAFRO is not included in iMCD. iMCD-TAFRO may be a part of iMCD.
2. In reference, the first name and family name were replaced for ref. number of 4 and 11.
3. The discussion part is too long and redundant. The author should shorten the length to around 2/3.
